# Does Constitutive Expression of Defense-Related Genes and Salicylic Acid Concentrations Correlate with Field Resistance of Potato to Black Scurf Disease?

**DOI:** 10.3390/bioengineering10111244

**Published:** 2023-10-24

**Authors:** Rita Zrenner, Franziska Genzel, Susanne Baldermann, Tiziana Guerra, Rita Grosch

**Affiliations:** 1Leibniz Institute of Vegetable and Ornamental Crops (IGZ) e.V., Theodor-Echtermeyer-Weg 1, 14979 Grossbeeren, Germany; f.genzel@fz-juelich.de (F.G.); tiziana.guerra@fu-berlin.de (T.G.); grosch@igzev.de (R.G.); 2Bioinformatics, Institute of Bio- and Geosciences, Forschungszentrum Jülich GmbH, Wilhelm-Johnen-Straße, 52428 Jülich, Germany; 3Faculty of Life Sciences: Food, Nutrition & Health, University Bayreuth, Fritz-Hornschuch-Straße 13, 95326 Kulmbach, Germany; susanne.baldermann@uni-bayreuth.de; 4Institute of Biology, Freie Universität Berlin, Königin-Luise-Str. 1-3, 14195 Berlin, Germany

**Keywords:** black scurf disease, defense-related gene expression, salicylic acid, *Rhizoctonia solani*, *Solanum tuberosum*

## Abstract

Black scurf disease on potato caused by *Rhizoctonia solani* AG3 occurs worldwide and is difficult to control. The use of potato cultivars resistant to black scurf disease could be part of an integrated control strategy. Currently, the degree of resistance is based on symptom assessment in the field, but molecular measures could provide a more efficient screening method. We hypothesized that the degree of field resistance to black scurf disease in potato cultivars is associated with defense-related gene expression levels and salicylic acid (SA) concentration. Cultivars with a moderate and severe appearance of disease symptoms on tubers were selected and cultivated in the same field. In addition, experiments were conducted under controlled conditions in an axenic in vitro culture and in a sand culture to analyze the constitutive expression of defense-related genes and SA concentration. The more resistant cultivars did not show significantly higher constitutive expression levels of defense-related genes. Moreover, the level of free SA was increased in the more resistant cultivars only in the roots of the plantlets grown in the sand culture. These results indicate that neither expression levels of defense-related genes nor the amount of SA in potato plants can be used as reliable predictors of the field resistance of potato genotypes to black scurf disease.

## 1. Introduction

The soil-borne fungus *Rhizoctonia solani* Kühn (teleomorph *Thanatephorus cucumeris* (Frank) Donk) is responsible for serious diseases in many crop plants worldwide. The fungus is a common and important pathogen in potato (*Solanum tuberosum* L.) that infects all belowground plant parts resulting in stem canker and black scurf disease [1,2,3]. The most obvious symptoms of Rhizoctonia diseases can be seen at harvest when dark sclerotia cover the maturing tubers (black scurf), thus leading to the reduced quality and marketability of the tubers with marketable yield losses approaching 10% to 50% [4,5,6].

The pathogen *R. solani* is a species complex and is categorized into anastomosis groups (AGs) based on hyphal anastomosis between isolates belonging to the same AG. Individual AGs and subgroups within an AG are associated with a particular host plant. Several studies confirm that isolates of AG3, or more specifically AG3PT (the potato type), are predominantly associated with Rhizoctonia disease in potato [7,8,9]. Recent control strategies are mainly dependent on chemical fungicides that are not able to sufficiently suppress the pathogen in potato [2,10,11], and efficient methods in practice are still lacking. The cultivation of Rhizoctonia-tolerant or -resistant potato cultivars would be a sustainable disease-management strategy. Based on the observation of differences in disease severity comparing potato cultivars, resistance to black scurf seems to be a quantitative trait [7,12,13]. Knowledge of the mechanisms underlying the potato–*R. solani* interaction can contribute to the development of screening methods for the selection of resistant potato genotypes.

Various studies reported about the role of plant innate immunity against Rhizoctonia diseases [14,15]. Zhao and coworkers [16] underline that a response to the necrotrophic pathogen *R. solani* in rice is a combination of general defense responses observed against different pathogens. Some processes like the production of reactive oxygen species (ROS), redox regulation or signal transduction were involved in the defense response against *R. solani* [14,17,18]. Meanwhile, various genes that confer resistance to *R. solani* were found especially in rice, such as genes that encoding-pathogenesis-related proteins, enzymes in the glycolytic and phenylpropanoid pathway or hormone-related proteins/enzymes [14,19,20]. Less is known about the defense mechanisms and pathways which are crucial for a manifestation of a high resistance level in potato cultivars and their inheritance [21,22]. Lehtonen and coworkers [23] found significant changes in gene expression patterns, and Hejazi and coworkers [24] additionally reported that antioxidant enzyme activities represent molecular and physiological events in infected potato sprouts. By analyzing antioxidant enzyme activity as well as biomass growth parameters, Soheili-Moghaddam and coworkers [25] revealed the important relationship between resistance and *R. solani* in potatoes.

Responses to (a)biotic factors are regulated by an array of signal transduction pathways within which phytohormones play a pivotal role [26,27,28]. The salicylic acid (SA) pathway is known to affect various plant processes including the induction of resistance especially more often to biotrophic and hemibiotrophic pathogens [29,30]. However, it was also shown that SA signaling plays a role in the response to necrotrophic pathogens as shown in oilseed rape and in tomato [31,32], thus resulting in a systemic defense response by the expression of defense-related genes and finally the production of pathogenesis-related (PR) proteins, phytoalexins and the strengthening of cell walls [33]. Several common pathogenesis-related genes like *PR1*, 1,3-β-glucanase (*PR2*) and chitinase (*PR3*) are associated with SA pathways and were upregulated in roots and sprouts in a susceptible potato cultivar as an early event in response to *R. solani* AG3 [34,35]. Glazebrook [36] reported about a positive correlation between endogenous levels of SA in plants and their resistance level to pathogens. Studies of De Vleesschauwer and coworkers [37] and Denancé and coworkers [38] showed as well that plants with a reduced ability to accumulate SA in tissues are more susceptible to root infection. A more robust and faster response to pathogen attack accompanied with SA accumulation was observed in Arabidopsis plants by Jung and coworkers [39]. Systemic SA accumulation in above-ground plant tissues can be induced by soil-borne pathogens [40]. However, only few studies were focused on the mechanisms of plant defense in roots, which are used for the invasion of the plant by a number of soilborne pathogens [41,42]. The pathogen *R. solani* AG3 infects not only the potato sprouts at an early plant stage but also the roots of the potato [34]. The necrotrophic lifestyle is observed especially on the emerging sprouts which are showing necrotic lesions. Afterwards, the pathogen is associated with belowground plant tissue during the whole growing period and appears as sclerotia on tubers at harvest. This raised up the question whether the observed differences in severity of black scurf disease between potato genotypes are related to different endogenous SA level in the plants.

Therefore, the objective of this study was to evaluate whether molecular tools that are useful for differentiating the resistance level of potato cultivars to black scurf disease could be identified prior to infestation with the pathogen. The following hypotheses were tested: (i) Is the quantitative resistance level positively related to the constitutive expression levels of relevant plant-defense-related genes and genes with relevance for the SA biosynthesis pathway? (ii) Does the constitutive expression level of these genes correlate with the amount of SA in potato tissue? (iii) Does a high expression level of the genes and the amount of SA reduce the severity of black scurf symptoms? For this purpose, various potato cultivars with different degrees of black scurf disease severity observed in the field were selected, and the resistance level was assessed at the same field site. A quantitative PCR (qPCR) was used for the evaluation of gene expression levels in the roots and shoots of these selected potato cultivars grown under optimal growth conditions in an axenic in vitro culture and in growth chambers in a sand culture in the absence of the pathogen. In addition, SA concentrations in the roots were measured.

## 2. Materials and Methods

### 2.1. Field Experiment

The potato cultivars ‘Granola’, ‘Lolita’, ‘Troja’, ‘Salute’, ‘Arkula’, ‘Jasia’, ‘Skonto’ and ‘Bonza’ showing differences in Rhizoctonia disease severity (DS) in the field were used in this study.

The rating of the resistance level of the selected cultivars to black scurf took place at the same field site (Großbeeren, Germany, 52°33′ N, 13°22′ E). The used field site (soil type: Diluvial sand) was naturally infested by *R. solani* based on the occurrence of black scurf disease on the harvested tubers of a previously cultivated potato crop. Plantlets from a tissue culture (see below) were used to make sure that the plant material is not contaminated with any pathogens at planting. Acclimatized plantlets were planted in plots (2 m × 2 m, 14 plantlets per plot). The spacing within the row amounted to 0.3 m and 0.65 m between rows. Before planting, the mineral content of the soil was adapted by fertilization with 160 kg/ha nitrogen and 300 kg/ha potassium (half of the amount at planting and the other half four weeks after planting). Each treatment included four plots arranged in a randomized block design. Irrigation was performed as required. Phytosanitary measures included weed removal by hand and the application of fungicides against *Phytophthora infestans* when needed. Four weeks after haulm death, all tubers were harvested, washed and graded according to their size for further examination.

Black scurf DS was assessed based on the percentage infestation of tubers with Rhizoctonia sclerotia by using the following scale from 1 to 5:1—without sclerotia, 2—<1%, 3—1–5%, 4—5–10% and 5—>10%. The average DS of black scurf from 120 randomly selected tubers was calculated (30 tubers per replicate).

### 2.2. Preparing Plantlets for Field Experiment

Since conventional seed tubers can already be latently infested with *R. solani* or other pathogens, in vitro cultured potato plantlets of the cultivars were used to assure working with pathogen-free plant material. The plantlets were grown in sterile plastic boxes (Sterivent high container, DUCHEFA; Haarlem, The Netherlands) in a Murashige and Skoog medium including Vitamins and an MES-buffer (DUCHEFA) with 2% sucrose at pH 5.8 and 0.8% PlantAgar (DUCHEFA) in a cultivator at (22)25(22) °C/20 °C day/night, (2)12(2) h/8 h day/night, with 100 μmol m^−2^ s^−1^ light. The plantlets were transferred into multitrays filled with a mixture (1:2) of quartz sand (Euroquarz; Dorsten, Germany) and substrate (Einheitserde Classic-Substrat Pikier, Einheitserdewerke Werkverband e.V.; Sinntal-Altengronau, Germany), covered with a lid and acclimatized for 4 days to autotrophic conditions in the greenhouse. Afterwards, plants were cultivated in the greenhouse under an average minimum temperature of 16.8 °C and an average maximum temperature of 28.3 °C for a further two weeks. The plants were poured with B’cuzz Hydro A + B nutrient solution (Atami B.V.; Rosmalen, The Netherlands), which had been adjusted to an EC of 2.1 dS m^−1^ and a pH of 5.8 by adding 4 M HNO_3_ as required.

### 2.3. In Vitro and Sand Culture Experiments

For the analysis of the constitutive expression level of defense associated genes and salicylic acid (SA) content, in vitro cultured potato plantlets of all the used cultivars were grown as described above. For the in vitro culture experiments, five plantlets of each cultivar were cultured together in sterile plastic boxes, and the roots and shoots were harvested separately 14 days after cutting a fresh tip with 3 leaves. Each replicate included the material of five plantlets from different boxes and was shock-frozen in liquid nitrogen and stored at −80 °C.

For the sand culture experiments, plantlets grown as described above were transferred in individual pots of 10 × 10 × 11 cm filled with quartz sand, covered with a lid and acclimatized for 4 days to autotrophic conditions in a growth chamber (York; Mannheim, Germany) under the following conditions: 18/15 °C day/night temperature, with 16 h/8 h day/night cycle, 400 μmol m^−2^ s^−1^ light and a relative humidity of 80%. The plants were fertilized with a B’cuzz Hydro A + B nutrient solution (Atami B.V.) adjusted to an EC of 2.1 dS m^−1^ and a pH of 5.8. After 24 days, three plants per cultivar and replicate were pooled, and subsequently, root and shoot samples were separately shock-frozen in liquid nitrogen and stored at −80 °C until use for the analysis of the expression of defense-related genes and the quantification of the SA concentration.

### 2.4. Expression Analysis of Defense-Related Genes in Potato Tissue

Analyses of the transcript level of the defense-related genes *PR1*, *PR2*, *PR3*, *PR6*, *PR10* and genes *PAL* and *ICS* were carried out by using quantitative reverse transcription polymerase chain reaction (qPCR) with oligonucleotide primer sets tested for reliable amplifications with efficiencies close to 2 (Appendix A).

The total RNA was extracted from 70–90 mg of ground shoot or root material by using an innuPREP Plant RNA Kit (Analytik Jena; Jena, Germany), and the quantity and quality of the RNA was checked with the bioanalyzer (Agilent Technologies Deutschland GmbH; Waldbronn, Germany). Single-stranded cDNA synthesis of 1 µg of the total RNA using an iScript cDNA Synthesis Kit (Bio-Rad Laboratories GmbH; Feldkirchen, Germany) in a 25 µL reaction was performed following the instructions of the manufacturer. A qPCR was performed by using 96-well reaction plates and Thermal Cycler CFX96 C1000 Touch (Bio-Rad) with the thermal profile of 95 °C for 5 min, 40 cycles of 95 °C for 15 s and 60 °C for 1 min, followed by a dsDNA melting curve analysis. Each reaction of 10 µL volume contained 200 nM of each primer, 3 µL of cDNA (1:10) and 5 µL of the Sensi Fast SYBR NO ROX Kit (Bioline GmbH; Luckenwalde, Germany). Data collection and analysis was performed with CFX Manager Software 3.0 (Bio-Rad). The respective biological replicates were measured in technical triplicates including nontemplate controls. Relative transcript levels were normalized on the basis of the expression of the invariant control actin (*ACT*). ΔCq was calculated as the difference between the control and target products (ΔCq = Cq*_ACT_* − Cq*_gene_*). Data were collected and compiled by using CFX Manager Software 3.0 (Bio-Rad Laboratories GmbH).

### 2.5. Determination of Salicylic Acid

The amount of SA was determined in the same shoot or root material used for the analysis of defense-related genes. The extraction of SA was carried out as previously described [43,44] with slight modifications: 100 mg (200 mg) of homogenized frozen leaf (root) material was extracted with 70% methanol and 90% methanol for 1 h at 65 °C. A total of 100 ng of SA-d4 (Sigma; Darmstadt, Germany) was added as an internal standard. The extracts were evaporated under N_2,_ and samples were resuspended in 5% TCA. The solution was partitioned against cyclohexane/ethyl acetate (1:1) two times, and the upper organic phase was evaporated. The residual sample, containing free SA, was dissolved in 80% formic acid/20% acetonitrile. The aqueous phase was acidified with one volume formic acid and incubated at 80 °C for 1 h. The solution was partitioned against cyclohexane/ethyl acetate (1:1) two times, and the upper organic phase was evaporated. The residual sample was dissolved in 80% formic acid/20% acetonitrile (contains conjugated SA) and analyzed by using an UHPLC-system (Agilent Technologies; Waldbronn, Germany) coupled to an Agilent 6530 QToF LC-MS (Agilent Technologies) as described [45].

### 2.6. Statistical Analyses

All statistical analyses were carried out by using the STATISTICA software package version 12.0 (StatSoft Inc.; Tulsa, OK, USA). The data regarding black scurf DS revealed in the field experiment was analyzed by a one-way ANOVA (factors: cultivar) combined with an LSD test (*p* ≤ 0.05) to evaluate the differences between the cultivars. Defense gene transcription data and SA contents were analyzed by a one-way ANOVA (*p* ≤ 0.05), and mean expression values and SA contents between the cultivars were tested by using a Tukey HSD test (*p* ≤ 0.05). Furthermore, Student’s *t*-tests were performed to assess significance among the two groups showing low or high disease severity.

## 3. Results

### 3.1. Field Experiment to Evaluate Disease Severity (DS) of Black Scurf on Potato Cultivars

The black scurf DS assessed on harvested potato tubers in the field experiment showed that the DS differed significantly depending on the cultivar (ANOVA, *p* ≤ 0.0000). The cultivars ‘Granola’, ‘Lolita’, ‘Troja’ and ‘Salute’ do not differ significantly in the DS of black scurf as well as the DS between the cultivars ‘Jasia’, ‘Skonto’ and ‘Bonza’ (Figure 1). A significantly lower percentage of sclerotia infestation on tubers was observed for the cultivars ‘Granola’, ‘Lolita’, ‘Troja’ and ‘Salute’ compared to the cultivars ‘Arkula’, ‘Jasia’, ‘Skonto’ and ‘Bonza’. No significant differences in the DS of black scurf symptoms on potato tubers (*p* ≤ 0.05) were revealed between the cultivars ‘Jasia’, ‘Skonto’ and ‘Bonza’ (Figure 1). ‘Arkula’ was the cultivar with the highest DS, which also differed from that of the cultivars ‘Skonto’ and ‘Bonza’. Consequently, in the further course of the analyses, the cultivars were divided into the groups DS low (‘Granola’, ‘Lolita’, ‘Troja’ and ‘Salute’) and DS high (‘Arkula’, ‘Jasia’, ‘Skonto’ and ‘Bonza’).

### 3.2. Expression of Defense-Related Genes in Potato Cultivars

For analyses of the constitutive expression levels of defense-associated genes, two different plant cultivation systems were chosen, which allow for the study of absolutely pathogen-free plant material: an axenic in vitro culture system used to maintain pathogen-free breeding lines, and a sand culture of pathogen-free plantlets under highly controlled conditions in a growth chamber. In order to distinguish the tissues which are targeted by the pathogen, the shoot and root material was analyzed separately.

#### 3.2.1. In Vitro Culture

Expression analyses revealed a high variability between the cultivars and tissues, with highest differences in the average expression levels of *PR1* and *PR2* in the shoots (Table 1) and *PR2* and *PR10* in the roots of the in vitro grown cultivars (Table 2).

When comparing the average expression levels between the two tissue types, *PR1* expression was much higher in the shoots and *PR2* levels were much higher in the roots. When comparing the average expression levels between the DS groups, a significantly lower expression level of *PR1*, *PR2* and *PR10* was present in the shoots of cultivars of the DS group low (‘Granola’, ‘Lolita’, ‘Troja’ and ‘Salute’) than in the DS group high (‘Arkula’, ‘Jasia’, ‘Skonto’ and ‘Bonza’). In contrast, average expression of *PR6* was significantly higher in the shoots of the DS group low cultivars. The comparison of the expression levels in the roots of the DS group low with DS group high showed almost no significant differences. Only *PR6* expression showed a significantly higher level in the roots of the cultivars with a low DS (Table 2).

In addition to *PR* gene expression, two other defense-related genes with relevance in SA biosynthesis were analyzed. A high variability between cultivars was detected in the expression of phenylalanine ammonia-lyase (*PAL*), while the expression of isochorismate synthase (*ICS*) did not show this variability (Table 3). A comparison of the expression levels between the two tissue types revealed no differences. When comparing the average expression levels between the DS groups, a lower expression level of *ICS* was present only in the shoots of the cultivars of the DS group low (‘Granola’, ‘Lolita’, ‘Troja’ and ‘Salute’) than in the DS group high (‘Arkula’, ‘Jasia’, ‘Skonto’ and ‘Bonza’).

#### 3.2.2. Sand Culture

Expression analyses of the sand-grown cultivars did not show high variability between cultivars and tissues. When comparing the average expression levels between the two tissue types, again *PR1* expression was higher in the shoots and *PR2* levels were higher in the roots (Table 4 and Table 5). When comparing the average expression levels between the DS groups, a significantly lower expression level of *PR1*, *PR2* and *PR6* was present in the shoots of the cultivars of the DS group low (‘Granola’, ‘Lolita’, ‘Troja’ and ‘Salute’) than in the DS group high (‘Arkula’, ‘Jasia’, ‘Skonto’ and ‘Bonza’) (Table 5). The comparison of the average expression levels in the roots of the DS group low with DS group high showed a significantly lower expression of *PR1* and *PR10* in the cultivars with a low DS (Table 5).

The analysis of *PAL* and *ICS* expression in sand-grown plants showed again a high variability of *PAL* expression between the cultivars (Table 6). A comparison of the expression levels between the two tissue types revealed a lower expression of *ICS* in the roots. When comparing the average expression levels between the DS groups, no significant differences were found.

In summary of the expression analyses, a more or less clear variation between the cultivars was found in the roots and shoots of plants grown in an axenic in vitro culture and also in a pathogen-free sand culture. However, the severity of disease symptoms of the plants grown in the field could not be clearly deduced from this variation.

### 3.3. Salicylic Acid Concentration in Potato Cultivars

In addition to expression analyses of defense-related genes, the content of free SA and glycosylated SA was determined in the same shoot or root material. The content of both free and glycosylated SA show high variability between cultivars and potato tissue (Figure 2a,b). Comparing the content of SA in the shoots of potato, an analysis revealed a significantly lower level of free SA than of glycosylated SA in all the potato cultivars (Figure 2a). The comparison of the mean free SA content in the shoots showed significantly lower content on average in the cultivars with a low DS (*t*-test, *p* < 0.05).

In contrast, an analysis of the SA content in the roots of the potato indicated no differences in the free and glycosylated SA levels (Figure 2b). A comparison of the SA level on average between the DS groups revealed a higher level of free and glycosylated SA in the roots of potato cultivars in the DS group low than of the DS group high. However, statistical analysis showed that these differences were not significant (*t*-test, *p* < 0.05) because of a high variation in the SA level between the replicates.

## 4. Discussion

Suitable methods are needed for host-breeding programs that allow for the reliable assessment of the quantitative resistance level of black scurf disease in potato genotypes based on bioassays. Field screening for resistant genotypes requires considerable time and manpower. In addition, the level of quantitative resistance as observed for black scurf disease on potato is highly affected by the growing conditions in the field [7,12]. The objective of this study was to evaluate whether molecular tools can be used to assess the quantitative resistance level of potato to black scurf disease by using a bioassay. An available, more-efficient screening method would be advantageous only if it correlates with field screening trials. Hence, various potato cultivars that differed in their quantitative resistance level to black scurf disease based on previous field observations were initially exposed to the same field conditions. The results of a disease severity (DS) assessment confirmed that differences in the DS of black scurf disease exist between potato cultivars. Based on the observed level of DS, the cultivars were subdivided in two groups: cultivars with a lower DS (‘Granola’, ‘Lolita’, ‘Troja’ and ‘Salute’) and higher DS (‘Arkula’, ‘Jasia’, ‘Skonto’ and ‘Bonza’) in the black scurf DS. These cultivars were used for an analysis of the expression levels of selected plant-defense-related genes associated with the SA pathway in vitro and sand culture experiments.

The genes studied here are known to be pathogenesis related and are in many cases induced by inoculation with the fungus [24,34,36]. Here it was hypothesized that the quantitative resistance level of potato cultivars to black scurf disease might be positively related to constitutive expression levels of plant-defense-related genes. For example, the gene products of *PR2* and *PR3* result in the production of hydrolytic enzymes that can immediately act against attacking fungi, thus reducing pathogen pressure. PR2 belongs to the group of β-1,3-glucanases that can either directly impair the growth of a fungus by hydrolyzing glucans of the fungal cell wall or indirectly through the released short glucan fragments from pathogen cell walls that can be recognized by plants, inducing further defense responses [46,47]. PR3 represents chitinases, which have been shown to inhibit the growth of different fungi in vitro either alone or in combination with β-1,3-glucanase [48]. Furthermore, previous results suggested that the constitutive expression of *PR* genes in leaves may contribute to nonspecific resistance to *Phythophthora infestans* in *Solanum* species [49] or is likely responsible for a large part of the partial resistance in rice against *Magnaporthe oryzae* [50]. Here, the average expression level of pathogenesis-related genes was assessed in shoots and roots at two different growth conditions. In both kinds of experiments, the constitutive expression level of *PR1*, *PR3* and *PR6* genes was higher in the shoots than in the roots whereas the *PR2* gene was more highly expressed in the roots. However, with the results presented, the hypothesis that the resistance level of potato cultivars to black scurf disease is positively related to the constitutive expression level of the studied defense-related genes must be rejected. Considering the DS groups, none of the studied genes were constitutively more highly expressed in the DS group low. More comprehensive analyses at the transcriptome level are needed to assess whether specific transcriptome patterns at early growth stages under strictly controlled conditions can be linked to the level of field resistance of potato cultivars to black scurf disease. Thus, a simple screening method for assessing the level of resistance of potato cultivars may have yet to be found.

Although the plants were not attacked by pathogens and kept under controlled growth conditions, a high variation in the average constitutive expression level of the studied plant-defense-related genes were observed within the DS groups. The genes *PR1*, *PR2*, *PR3*, *PR6* and *PR10* were selected, some of which are directly associated with the SA signaling pathway (*PR1*, *PR2* and *PR10*), while others are primarily induced by jasmonic acid (JA) (*PR3* and *PR6*). Evidence that *R. solani* induces SA-mediated signaling pathways leading to a higher expression of, for instance, the *PR1* gene was found in rice, among others [16]. Hence, this study focuses on the role of defense-related genes associated with the SA signaling pathway. Studies of Shah and Zeier [51] demonstrated that a higher level of SA is associated with an enhanced expression of the *PR1* gene and requires an enhanced expression of the *ICS* gene, which is important for SA biosynthesis in plant tissue. However, such relationships could not be found in potato plants grown in highly controlled conditions. A higher endogenous level of SA did not correlate with an increased expression level of genes associated with the SA signaling pathway. Hence, the hypothesis that the constitutive expression level of the studied genes and genes of relevance for the SA biosynthesis pathway (*PAL*, *ICS*) correlate with the amount of SA in potato tissue must be rejected, at least for plants that are not infested by pathogens.

Classically, the jasmonic acid (JA) pathway is involved primary in the defense response to necrotrophic pathogens [52], as also shown against *R. solani* in rice [53]. However, some studies demonstrated that resistance to necrotrophic pathogens also requires the signaling of SA [54,55]. The importance of intact SA signaling for potato defense against the necrotrophic fungal pathogen *Alternaria solani* was shown by Brouwer et al. [56]. The pathogen *R. solani* is described as a pathogen with a necrotrophic lifestyle; however, the possibility of a combination of necrotrophic and hemibiotrophic behavior is assumed in potato [35] and as well in rice [57]. In nature, plants can be impacted by both biotrophic and necrotrophic pathogens, and therefore it is crucial for plants to have elevated levels of both JA and SA [58]. It was hypothesized that the quantitative resistance level of potato cultivars to black scurf disease is positively related to the amount of SA in the tissue and can already be inferred from plants grown under controlled conditions in the absence of the pathogen. The amount of SA measured varied by cultivar. However, no consistent separation was evident in the low and high DS groups. Thus, it can be concluded that the severity of black scurf disease symptoms on the tubers of plants grown in the field could not be clearly inferred from this variation. It remains to be tested whether the differential induction of SA content in roots by the presence of the pathogen can reveal such a relationship. However, this would be far from a simple screening procedure to assess the resistance level of potato cultivars in early stages. Further analyses of relevant enzyme activities [25] or unbiased analyses at the proteome or metabolome level could lead to specific candidates that would make it possible in the future to establish a link with the degree of field resistance of potato varieties to black scurf disease.

## 5. Conclusions

Field experiments showed the separation of potato cultivars in a low and high disease severity (DS) group when challenged with *Rhizoctonia solani* AG3. The more resistant cultivars did not show significantly higher constitutive expression levels of defense-related genes. Only free SA was increased in the roots of more resistant cultivars grown in a sand culture. These results indicate that neither the expression levels of the analyzed defense-related genes nor the amount of SA in potato plants can be used as reliable predictors of the field resistance of potato to black scurf disease.

## Figures and Tables

**Figure 1 bioengineering-10-01244-f001:**
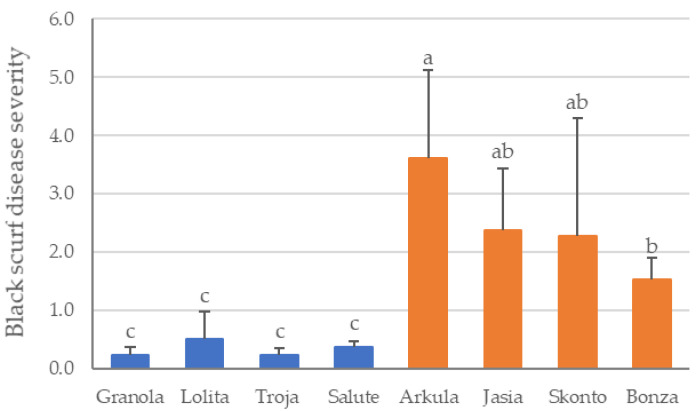
Disease severity of black scurf on potato cultivars in the field. Mean value and standard deviation (n = 4) are shown for disease severity group low (‘Granola’, ‘Lolita’, ‘Troja’ and ‘Salute’) in blue and for disease severity group high (‘Arkula’, ‘Jasia’, ‘Skonto’ and ‘Bonza’) in orange. DS was assessed as percentage infestation of tubers with sclerotia using scale from 1 to 5 with 1: without sclerotia, 2: <1%, 3: 1–5%, 4: 5–10% and 5: >10%. The average DS of black scurf from 120 randomly selected tubers was calculated (30 tubers per replicate). Different letters indicate significance among the cultivars (*p* < 0.05).

**Figure 2 bioengineering-10-01244-f002:**
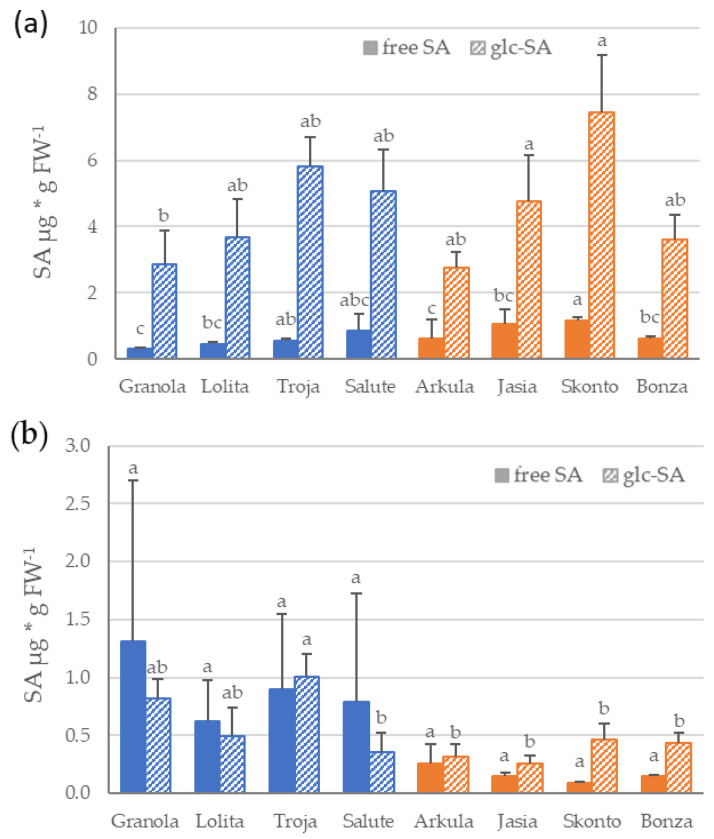
Salicylic acid (SA) content in shoots (**a**) and roots (**b**) of potato cultivars grown in sand culture. Free SA and glycosylated SA (glc-SA) levels were measured separately. Mean value and standard deviation (n = 4) are shown for disease severity group low (‘Granola’, ‘Lolita’, ‘Troja’ and ‘Salute’) in blue, and for disease severity group high (‘Arkula’, ‘Jasia’, ‘Skonto’ and ‘Bonza’) in orange. Different letters indicate significance among the cultivars within each metabolite analysis (*p* < 0.05); Student’s *t*-test shows significance of free SA content among the disease severity groups low and high (*p* < 0.05). SA, salicylic acid; glc-SA, glycosylated salicylic acid.

**Table 1 bioengineering-10-01244-t001:** Relative expression levels of *PR1*, *PR2*, *PR3*, *PR6* and *PR10* in shoots of potato cultivars grown in axenic in vitro culture. Student’s *t*-tests indicate significance among the disease severity groups low (‘Granola’, ‘Lolita’, ‘Troja’ and ‘Salute’) and high (‘Arkula’, ‘Jasia’, ‘Skonto’ and ‘Bonza’) within each gene-expression analysis (n = 5).

Cultivar	DS	*PR1*			*PR2*			*PR3*			*PR6*			*PR10*		
		MV	*S*	SD	MV	*S*	SD	MV	*S*	SD	MV	*S*	SD	MV	*S*	SD
Granola	low	5.3	abc	1.63	2.3	abc	0.62	4.4	a	0.97	6.2	abc	1.35	5.9	b	0.29
Lolita	low	3.1	bcd	1.77	−2.3	d	0.52	2.4	a	1.80	6.2	abc	1.26	5.4	b	0.25
Troja	low	2.2	cd	0.98	−1.7	d	1.26	3.8	a	0.84	6.4	ab	1.01	5.6	b	0.68
Salute	low	0.3	d	0.29	−0.4	cd	0.67	4.1	a	0.35	6.4	abc	1.32	5.8	b	0.20
Arkula	high	6.3	ab	3.68	0.5	bcd	1.76	4.6	a	2.68	6.8	a	1.16	5.1	b	0.73
Jasia	high	4.3	a	0.58	2.8	a	0.70	3.5	a	1.87	4.4	d	0.87	6.7	a	0.43
Skonto	high	4.4	abc	1.13	2.6	ab	2.50	4.0	a	1.66	4.1	bcd	1.03	6.5	b	1.06
Bonza	high	8.0	abc	2.11	4.6	ab	1.49	5.4	a	1.00	3.9	cd	0.70	9.0	b	1.70
*t*-test		*			*			ns			*			*		

*, significantly different (*t*-test, *p* < 0.05); ns, no significance among both DS groups. *S*, significance, different letters indicate significance among cultivars within each gene-expression analysis (*p* < 0.05); DS, disease severity; MV, mean value; SD, standard deviation.

**Table 2 bioengineering-10-01244-t002:** Relative expression levels of *PR1*, *PR2*, *PR3*, *PR6* and *PR10* in roots of potato cultivars grown in axenic in vitro culture. Student’s *t*-tests indicate significance among the disease severity groups low (‘Granola’, ‘Lolita’, ‘Troja’ and ‘Salute’) and high (‘Arkula’, ‘Jasia’, ‘Skonto’ and ‘Bonza’) within each gene-expression analysis (n = 5).

Cultivar	DS	*PR1*			*PR2*			*PR3*			*PR6*			*PR10*		
		MV	*S*	SD	SD	*S*	SD	MV	*S*	SD	MV	*S*	SD	MV	*S*	SD
Granola	low	1.7	a	0.71	0.90	b	2.54	3.4	a	0.50	3.4	a	0.90	0.4	b	1.42
Lolita	low	−0.6	a	2.40	0.76	a	0.56	2.8	a	1.33	3.6	a	0.76	10.3	a	0.76
Troja	low	−0.2	a	1.19	0.99	a	1.15	3.0	a	1.31	3.2	a	0.99	8.9	a	0.79
Salute	low	−0.2	a	1.42	0.84	a	0.29	2.9	a	0.66	3.1	a	0.84	9.5	a	0.84
Arkula	high	1.2	a	0.56	0.98	b	2.20	4.2	a	1.67	3.4	a	0.98	0.1	b	1.02
Jasia	high	0.7	a	2.08	0.71	a	0.70	3.1	a	3.88	2.5	a	0.71	9.4	a	1.46
Skonto	high	−1.0	a	1.12	1.30	a	0.33	3.5	a	0.40	2.4	a	1.30	8.8	a	0.56
Bonza	high	0.3	a	1.88	0.62	a	0.23	3.2	a	1.36	1.9	a	0.62	11.0	a	1.96
*t*-test		ns			ns			ns			*			ns		

*, significantly different (*t*-test, *p* < 0.05); ns, no significance among both DS groups. *S*, significance, different letters indicate significance among cultivars within each gene-expression analysis (*p* < 0.05); DS, disease severity; MV, mean value; SD, standard deviation.

**Table 3 bioengineering-10-01244-t003:** Relative expression levels of *PAL* and *ICS* in shoots and roots of potato cultivars grown in axenic in vitro culture. Student’s *t*-tests indicate significance among the disease severity groups low (‘Granola’, ‘Lolita’, ‘Troja’ and ‘Salute’) and high (‘Arkula’, ‘Jasia’, ‘Skonto’ and ‘Bonza’) within each gene-expression analysis (n = 5).

		Shoots						Roots					
Cultivar	DS	*PAL*			*ICS*			*PAL*			*ICS*		
		MV	*S*	SD	MV	*S*	SD	MV	*S*	SD	MV	*S*	SD
Granola	low	6.3	a	1.67	2.0	bc	0.38	8.2	a	0.70	1.4	bc	0.72
Lolita	low	1.7	b	0.73	1.1	c	0.18	1.7	b	0.74	1.0	bc	0.36
Troja	low	2.1	b	0.76	1.4	c	0.73	1.6	b	0.55	2.2	ab	0.48
Salute	low	5.8	a	0.53	1.1	c	0.37	7.7	a	0.99	1.4	bc	0.19
Arkula	high	6.2	a	0.95	1.6	c	0.95	8.1	a	0.96	1.1	bc	0.88
Jasia	high	2.3	b	0.93	3.1	a	0.68	2.4	b	0.69	1.4	a	0.85
Skonto	high	1.8	b	0.85	1.5	ab	0.45	1.2	b	0.84	0.1	bc	0.25
Bonza	high	1.3	b	0.41	4.4	c	0.88	1.8	b	0.96	3.5	c	0.96
*t*-test		ns			*			ns			ns		

*, significantly different (*t*-test, *p* < 0.05); ns, no significance among both DS groups. *S*, significance, different letters indicate significance among cultivars within each gene-expression analysis (*p* < 0.05); DS, disease severity; MV, mean value; SD, standard deviation.

**Table 4 bioengineering-10-01244-t004:** Relative expression levels of *PR1*, *PR2*, *PR3*, *PR6* and *PR10* in shoots of potato cultivars grown in sand culture. Student’s *t*-tests indicate significance among the disease severity groups low (‘Granola’, ‘Lolita’, ‘Troja’ and ‘Salute’) and high (‘Arkula’, ‘Jasia’, ‘Skonto’ and ‘Bonza’) within each gene-expression analysis (n = 4).

Cultivar	DS	*PR1*			*PR2*			*PR3*			*PR6*			*PR10*		
		MV	*S*	SD	MV	*S*	SD	MV	*S*	SD	MV	*S*	SD	MV	*S*	SD
Granola	low	5.1	bc	1.14	3.2	a	1.31	7.8	a	1.74	6.7	b	1.05	8.3	a	1.42
Lolita	low	3.9	bc	1.00	1.6	a	0.89	7.0	a	1.23	6.1	b	0.80	8.1	a	1.03
Troja	low	5.9	abc	2.34	2.5	a	1.94	7.7	a	2.24	7.2	ab	2.13	8.5	a	1.78
Salute	low	3.4	c	0.87	2.9	a	1.02	7.7	a	1.40	6.6	b	1.63	8.0	a	0.84
Arkula	high	7.4	abc	0.40	3.3	a	1.04	7.6	a	1.43	6.9	b	1.76	8.1	a	1.32
Jasia	high	4.8	ab	0.63	2.4	a	1.01	8.1	a	0.75	8.1	ab	0.40	7.6	a	0.20
Skonto	high	9.3	bc	2.34	5.5	a	2.24	10.2	a	2.45	11.1	ab	2.02	10.0	a	2.14
Bonza	high	7.9	a	1.35	5.3	a	1.88	9.0	a	1.62	8.0	a	0.86	9.3	a	1.19
*t*-test		*			*			ns			*			ns		

*, significantly different (*t*-test, *p* < 0.05); ns, no significance among both DS groups. *S*, significance, different letters indicate significance among cultivars within each gene-expression analysis (*p* < 0.05); DS, disease severity; MV, mean value; SD, standard deviation.

**Table 5 bioengineering-10-01244-t005:** Relative expression levels of *PR1*, *PR2*, *PR3*, *PR6* and *PR10* in roots of potato cultivars grown in sand culture. Student’s *t*-tests indicate significance among the disease severity groups low (‘Granola’, ‘Lolita’, ‘Troja’ and ‘Salute’) and high (‘Arkula’, ‘Jasia’, ‘Skonto’ and ‘Bonza’) within each gene-expression analysis (n = 4).

Cultivar	DS	*PR1*			*PR2*			*PR3*			*PR6*			*PR10*		
		MV	*S*	SD	MV	*S*	SD	MV	*S*	SD	MV	*S*	SD	MV	*S*	SD
Granola	low	2.4	bc	0.30	6.1	ab	1.27	6.5	a	1.46	3.8	abc	1.04	9.4	ab	0.95
Lolita	low	1.7	bc	0.94	5.6	ab	1.76	4.1	a	0.78	3.9	abc	1.25	9.2	ab	0.57
Troja	low	2.1	bc	0.65	4.6	ab	1.62	4.5	a	0.25	2.1	c	0.91	8.4	b	0.44
Salute	low	1.3	c	1.55	3.8	b	2.07	3.4	a	0.30	2.7	bc	0.63	8.6	b	0.56
Arkula	high	1.9	bc	0.56	4.5	ab	0.86	4.6	a	1.40	2.9	abc	0.09	9.8	ab	1.37
Jasia	high	3.5	a	0.79	6.1	a	0.40	5.4	a	1.18	3.4	a	0.97	9.2	ab	0.88
Skonto	high	2.7	ab	1.28	5.9	ab	1.18	6.6	a	2.50	4.3	abc	0.88	11.4	ab	1.80
Bonza	high	4.9	bc	0.26	7.2	ab	0.37	5.3	a	0.30	4.9	ab	0.47	9.1	a	0.20
*t*-test		*			ns			ns			ns			*		

*, significantly different (*t*-test, *p* < 0.05); ns, no significance among both DS groups. *S*, significance, different letters indicate significance among cultivars within each gene-expression analysis (*p* < 0.05); DS, disease severity; MV, mean value; SD, standard deviation.

**Table 6 bioengineering-10-01244-t006:** Relative expression levels of *PAL* and *ICS* in shoots and roots of potato cultivars grown in sand culture. Student’s *t*-tests indicate significance among the disease severity groups low (‘Granola’, ‘Lolita’, ‘Troja’ and ‘Salute’) and high (‘Arkula’, ‘Jasia’, ‘Skonto’ and ‘Bonza’) within each gene-expression analysis (n = 4).

		Shoots						Roots					
Cultivar	DS	*PAL*			*ICS*			*PAL*			*ICS*		
		MV	*S*	SD	MV	*S*	SD	MV	*S*	SD	MV	*S*	SD
Granola	low	5.5	a	0.74	1.7	a	0.67	6.7	a	0.89	−1.7	a	1.45
Lolita	low	−2.6	b	0.45	1.2	a	0.63	−0.7	b	0.76	−2.2	a	0.35
Troja	low	−2.5	b	1.32	1.5	a	0.87	−1.3	b	1.38	−2.5	a	0.63
Salute	low	5.4	a	0.59	1.6	a	0.14	6.8	a	1.11	−1.4	a	0.60
Arkula	high	4.5	a	1.57	1.5	a	0.99	6.5	a	0.39	−0.6	a	1.49
Jasia	high	6.6	b	0.48	1.8	a	0.36	7.9	b	0.79	−2.3	a	1.25
Skonto	high	−1.1	a	1.21	1.8	a	0.66	−0.7	a	0.92	−0.6	a	1.01
Bonza	high	−2.0	b	1.01	1.9	a	0.82	−1.4	b	1.07	−2.0	a	0.40
*t*-test		ns			ns			ns			ns		

ns, no significance among both DS groups (*t*-test, *p* < 0.05). *S*, significance, different letters indicate significance among cultivars within each gene-expression analysis (*p* < 0.05); DS, disease severity; MV, mean value; SD, standard deviation.

## Data Availability

The data presented in this study are available from the corresponding author upon request.

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
