# Peer review of "Does Constitutive Expression of Defense-Related Genes and Salicylic Acid Concentrations Correlate with Field Resistance of Potato to Black Scurf Disease?"

_bioengineering, 2023, doi:10.3390/bioengineering10111244_

Round 1

Reviewer 1 Report

Comments and Suggestions for Authors

Dear authors
Happy day
The paper is in need for readjustment particularly for the statistical analysis part. The control is missing and used internally to calculate the severity of the disease. There is no need for doing additional experiments. I suggest reanalysis the data but include the control as a factor. Additionally use the cluster analysis including the control to show the different level of correlations.

I recommended reading this paper before doing the statistical analysis 
-"2008 Plant crude extracts could be the solution: extracts showing in vivo antitumorigenic activity. Pak. J. Pharm. Sci., 21(2): 159-171"

-"2006. Case-by-case study using antibiotic-EDTA combination to control Pseudomonas aeruginosa. Pak. J. Pharm. Sci., 19(3), 236-243"
and 

Additionally, there are some comments about the writing style which might improve the paper.
1- Kindly use abbreviations whenever a group of words is repeated. For example, "Black scurf disease" - You, kindly can use "BSD". But better avoid use abbreviation in the abstract.
2- The introduction part is too long and contain your own point of view. Kindly:
q- make it shorter
b-Move any opinion or personal comment to the dissolution part.
c- It might be better if you the lines from 105 to 117 to the discussion part with the proper modification. 
3- The material and methods us fine. 

4- The results: 
a-Kindly include the control even it has 0 result in your figure or in a separate table. Or. mention that, the control did not included in the statistical analysis because you use it to calculate the infection load. Or, at least show us the control results of the different experiments in one Table.
b- It might be a weak point in the paper that you did not use the control correctly in your data analysis, you use it internally only to show the severity of the infection. But the control is highly critical to prove that the factors your analysis are effective. Kindly, re-think about your statistical analysis.
b- in figure 1 there is no description for the meaning of the bar’s lines. and in figure 2 and 3 the colors are in-compatible.
5- I am not sure from that the author can concluded that "These results suggest that neither expression levels of the analyzed defense-related genes nor the amount of SA in potato plants can be used as reliable indicators of field resistance of potato to black scurf disease."

with my pleasure

Reviewer 2 Report

Comments and Suggestions for Authors

1. Improve the introduction section with recent studies, and write the major scope of this study. Move Table 1 into supplementary materials

2. Please add the significance letter in the correct place; it is not correct to mention it in a separate column.

3. If any of the presented information about primers and qPCR in potatoes (including reference gene Actin) was published, please provide references. This information has to be present in the Supplementary file with full details.

4. Why did authors focus only on SA, there are many disease resistance-related pathways

 5. Combine figure 2 and 3

6.  Figure 2 and 3 E error bars are very high, it looks like an experimental error  

7. The discussion section needs subsequent improvement with the latest references 

Comments on the Quality of English Language

Moderate editing of English language required

Round 2

Reviewer 1 Report

Comments and Suggestions for Authors

Dear authors

The paper has significantly improved.

No comments on the pdf file

with my pleasure

Reviewer 2 Report

Comments and Suggestions for Authors

I recommend the manuscript for publication to Bioengineering

Comments on the Quality of English Language

Minor editing of the English language required